# Deltamethrin-Evoked ER Stress Promotes Neuroinflammation in the Adult Mouse Hippocampus

**DOI:** 10.3390/cells11121961

**Published:** 2022-06-18

**Authors:** Muhammad M. Hossain, Abigail C. Toltin, Laura M. Gamba, Maria A. Molina

**Affiliations:** Department of Environmental Health Sciences, Robert Stempel College of Public Health & Social Work, Florida International University, Miami, FL 33199, USA; atoltin@fiu.edu (A.C.T.); lgamba@fiu.edu (L.M.G.); mamolina@fiu.edu (M.A.M.)

**Keywords:** ER stress, oxidative and nitrosative stress, neuroinflammation, deltamethrin, salubrinal

## Abstract

Endoplasmic reticulum (ER) stress and neuroinflammation are involved in the pathogenesis of many neurodegenerative disorders. Previously, we reported that exposure to pyrethroid insecticide deltamethrin causes hippocampal ER stress apoptosis, a reduction in neurogenesis, and learning deficits in adult male mice. Recently, we found that deltamethrin exposure also increases the markers of neuroinflammation in BV2 cells. Here, we investigated the potential mechanistic link between ER stress and neuroinflammation following exposure to deltamethrin. We found that repeated oral exposure to deltamethrin (3 mg/kg) for 30 days caused microglial activation and increased gene expressions and protein levels of TNF-α, IL-1β, IL-6, gp91^phox^, 4HNE, and iNOS in the hippocampus. These changes were preceded by the induction of ER stress as the protein levels of CHOP, ATF-4, and GRP78 were significantly increased in the hippocampus. To determine whether induction of ER stress triggers the inflammatory response, we performed an additional experiment with mouse microglial cell (MMC) line. MMCs were treated with 0–5 µM deltamethrin for 24–48 h in the presence or absence of salubrinal, a pharmacological inhibitor of the ER stress factor eIF2α. We found that salubrinal (50 µM) prevented deltamethrin-induced ER stress, as indicated by decreased levels of CHOP and ATF-4, and attenuated the levels of GSH, 4-HNE, gp91^phox^, iNOS, ROS, TNF-α, IL-1β, and IL-6 in MMCs. Together, these results demonstrate that exposure to deltamethrin leads to ER stress-mediated neuroinflammation, which may subsequently contribute to neurodegeneration and cognitive impairment in mice.

## 1. Introduction

The endoplasmic reticulum (ER) is a vital organelle that is responsible for protein synthesis, modifications, proper folding, and transportation across the cell [1,2,3], as well as maintaining calcium homeostasis for normal cell function [4]. Loss of integrity or perturbance of the ER functions leads to the accumulation of misfolded proteins or the disruption of calcium homeostasis, which initiate ER stress [1,5]. Under ER stress, cells activate the highly specific unfolded protein response (UPR) system to counteract the stress. The UPR promotes the refolding or degradation of misfolded proteins and inhibits protein synthesis. When robust, sustained levels of ER stress are sensed by cells, UPR function is impaired and the activation of inflammatory and apoptotic pathways is triggered [2,5]. Accumulations of misfolded proteins in the ERs of neurons and microglial cells are often found in the pathogenesis of several neurodegenerative diseases [2,6]. The UPR-induced inflammatory and apoptotic pathways contribute to neuroinflammation, a hallmark of neurodegenerative diseases. Consequently, several neurodegenerative diseases such as Alzheimer’s, Parkinson’s, amyotrophic lateral sclerosis, and multiple sclerosis display a phenomenon where pathologic inflammation in neurons leads to chronic and persistent neurodegeneration [7,8]. 

Exposure to environmental toxicants, including pesticides, has been shown to be a causative agent of ER stress and neuroinflammation associated with neurodegeneration and neurological diseases [2,9,10]. Pyrethroids are the group of insecticides most widely employed to control pests in public health, agriculture, and residential settings [11,12]. Increased pyrethroid exposure to the general public occurs through diet, dust, and aerosolized particles found in a variety of household applications [13,14]. Pyrethroid insecticides are divided into two groups, type-I and type-II, based on their toxicological and structural differences [12,15]. Among the two classes of pyrethroids, type-II pyrethroid deltamethrin has received a significant amount of attention, as it is a main active ingredient in many households insecticides, and poses a considerable risk to human health due to the lack of the enzyme carboxylesterase, which detoxifies most pyrethroids in humans [11,16]. Although deltamethrin induces neurotoxicity by interacting with sodium channels, other cellular mechanisms have also been implicated, including mitochondrial dysfunction, ER stress, and microglial activation [3,5,17].

In previous studies, we showed that pyrethroid exposure induced ER stress, leading to neurodegeneration [3,10,18]. Recently, we reported that pyrethroid insecticides activate microglia and increase the production of the pro-inflammatory cytokine tumor necrosis factor alpha (TNF-α) in a BV2 cells [3]. Here, we investigated whether exposure to deltamethrin causes neuroinflammation and oxidative/nitrosative stress in the hippocampus of adult mice. In addition, using a mouse microglial cell (MMC) line, we sought to identify whether the deltamethrin-induced neuroinflammation was caused by the ER stress pathways. Our data uncover that exposure to deltamethrin leads to oxidative/nitrosative stress and neuroinflammation, as indicated by decreased levels of glutathione (GSH) and increased levels of gp91^phox^, inducible nitric oxide synthase (iNOS), IL-1β (interleukin 1beta), TNF-α, and interleukin 6 (IL-6), respectively, in MMCs. Further, inhibition of ER stress factor eIF2α with salubrinal mitigates the effects of deltamethrin on the reduction in GSH levels and the induction of gp91^phox^, 4-HNE, iNOS, ROS, IL-1β, TNF-α, and IL-6 in MMCs cells, indicating that pyrethroid insecticides may cause neuroinflammation through the activation of ER stress pathways. 

## 2. Materials and Methods

### 2.1. Animals

Male C57BL/6J mice (10 weeks old) were obtained from Jackson Laboratory (Bar Harbor, ME, USA), and housed four animals/cage in a temperature (72 ± 1 °C) and humidity (45 ± 5%) controlled vivarium on a 12 h light–dark cycle, with free access to food and water. Experiments were conducted in accordance with the NIH Guideline for the Care and Use of Laboratory Animals and approved by the Intuitional Animal Care and Use Committee of Florida International University (IACUC-18-070). 

### 2.2. Treatment

Twenty (20) mice were randomly allocated into two groups: control (*n* = 10) and treatment (*n* = 10). Animals in the control group were given corn oil alone, and animals in the treatment group were given 3 mg/kg of deltamethrin in corn oil by oral gavage once every three days for thirty (30) days (Figure 1) as we previously reported [19]. Mice treated with deltamethrin did not display any typical signs and symptoms of toxicity, such as salivation, tremors, and/or weight loss, throughout the entire study. Dosing was administered in 3-day intervals based on a physiologically based pharmacokinetic study demonstrating that deltamethrin is almost eliminated within 48 h of oral administration [20]. Twenty-four hours after the last dose, 5 mice per group were sacrificed, and hippocampi were collected for gene and protein analysis and the remaining animals (*n* = 5/group) were perfused for immunohistochemical study (Figure 1). 

### 2.3. Cell Culture

The immortalized mouse microglial cell (MMC) line was used as an *in vitro* cell culture model for testing deltamethrin-induced neuroinflammatory responses [21]. MMCs were grown in DMEM/F12 (Cat #11320-033, Gibco, Life Technologies Corporation, Grand Island, NY, USA) media supplemented with 10% heat inactivated FBS, 2 mM L-glutamine, 1 mM sodium pyruvate, 1 mM non-essential amino acids, 50 IU penicillin, and 50 µg/ml streptomycin, and were maintained in a humidified atmosphere of 5% CO_2_ at 37 °C. Following 3 days culturing in a flask, the cells were resuspended in DMEM/F12 medium and then plated at 1 × 10^6^ cells/well into 6-well plates for Western blot assay and 2.5 × 10^3^/well into 96-well plates for immunofluorescence staining. A 10 mM deltamethrin stock solution was prepared in absolute ethanol (EtOH). Cells were treated with 0–5 µM of deltamethrin in the presence or absence of 50 μM salubrinal (Cat # SML0951, Sigma-Aldrich, Inc., St. Louis, MO, USA) and were then harvested at 24 h and 48 h of exposure. These concentrations of deltamethrin were selected as they did not cause cytotoxicity to MMCs (Appendix A). 

### 2.4. Reactive Oxygen Species (ROS) Assay

Generation of intracellular ROS was determined using the fluorescent dye H_2_DCFDA (Molecular Probe, Grand Island, NY, USA) as we previously described [22]. Briefly, after 12 h culturing, MMCs were incubated with 5 μM H_2_DCFDA at 37 °C for 30 min. Cells were washed twice with PBS and treated with delatamethrin with or without salubrinal. Then fluorescent intensity was measured at 2 h and 6 h on a SpectraMax M5 (Molecular Devices, LLC., San Jose, CA, USA) microplate reader to quantify the levels of intracellular ROS.

### 2.5. Intracellular Glutathione (GSH) Assay 

MMCs were seeded in 96-well plates and were treated with deltamethrin (0–5 µM) ± salubrinal (50 µM) for 24 and 48 h. Following the treatment, cells were harvested in lysis buffer and then the concentration of glutathione was measured using a glutathione assay kit (Cat #AB65322, Abcam, Cambridge, MA, USA), according to the manufacturer’s instructions. Briefly, samples and GSH standards were incubated with glutathione S-transferase (GHT) and monochlorobimane (MCB) for 60 min at 37 °C and then fluorescence was measured with excitation 380 nm and emission 461 nm using a SpectraMax M5 (Molecular Devices, LLC., San Jose, CA, USA) microplate reader. The concentration of GSH in the samples was calculated as nmol/mg protein. Protein content was determined using Pierce^TM^ BCA protein assay kit (Cat #23225, Thermo Fisher Scientific, Rockford, IL, USA).

### 2.6. RNA Isolation and Real-Time Quantitative Polymerase Chain Reaction (qPCR)

RNA isolation, cDNA synthesis, and Real-time qPCR were performed as we previously described [22,23,24]. Briefly, total RNA from hippocampal tissues and cells was extracted using Trizol/chloroform extraction and RLT buffer containing 1% β-Me, respectively, with Qiagen RNeasy^®^ mini kits (QIAGEN, Valencia, CA, USA), and then RNA was converted to cDNA using the First Strand Super Script cDNA synthesis kit (Invitrogen, San Diego, CA, USA). To determine mRNA expression, a Real-time qPCR was performed using iTag^TM^ Universal SYBR^®^ Green Suppermix (Cat #1725124, Bio-Rad, Hercules, CA, USA) with the ABI 7900HT (Applied Biosystems, Foster City, CA, USA) detection system. The disassociation and melting curves were obtained to confirm that a single peak was generated by each PCR product. The relative changes in mRNA expression were quantified using the ΔΔCt method with the threshold cycle (Ct) value of GAPDH as reference. The primer sequences for target mRNA are presented in Table 1.

### 2.7. Western Blot Analysis

Extraction of protein and Western blot (WB) analysis were performed as we previously described [5,19]. After quantification of protein concentration using Pierce BCA Protein Assay Kit (Cat #23227, Thermo Fisher Scientific, Rockford, IL, USA), 20 µg of protein/lane was separated on 4–12% Bis-Tris gels. Separated protein bands were then transferred to PVDF membranes and blocked in 7.5% non-fat milk for 1 h. The membranes were then incubated overnight at 4 °C with anti-GRP78 (1:1000; Cat #3177, Cell Signaling, Danvers, MA, USA), anti-CHOP (1:500, Cat #SC575, Santa Cruz, CA, USA), anti-ATF-4 (1:1000, Cat #11815, Cell Signaling, Danvers, MA, USA), anti-TNF-α (1:1000, Cat #AB1793, Abcam, Cambridge, MA, USA), anti-IL-1β (1:1000, Cat #AB1413-I, Millipore sigma, Burlington, MA, USA), anti-IL-6 (1:1000, Cat #AB229381, Abcam, Cambridge, MA, USA), anti-gp91^phox^ (1:1000, Cat #SC27635, Santa Cruz, CA, USA), 4-HNE (1:1000, cat #MA5-27570, Thermo Fisher Scientific, Rockford, IL, USA), and anti-iNOS (1:100, Cat #SC7271, Santa Cruz, CA, USA) primary antibodies. Membranes were rinsed with TTBS and incubated with their appropriate secondary antibodies for 1 h. The proteins were detected with SuperSignal^®^ West Dura Extended Duration Substrate (Cat #34076, Thermo Fisher Scientific, Rockford, IL, USA) using a Bio-Rad (Hercules, CA, USA) imaging system, and quantified with AlphaView 3.5.0 software (ProteinSimple, San Jose, CA, USA). Subsequently, membranes were washed, stripped in stripping buffer (Cat #46430, Thermo Fisher Scientific, Rockford, IL, USA), and re-probed with β-actin (1:4000, Cat #A5441, Sigma-Aldrich, Inc., St. Louis, MO, USA), or α-tubulin antibody (1:4000, Cat #T5168, Sigma-Aldrich, Inc., St. Louis, MO, USA) to ensure that equal protein was loaded in each lane.

### 2.8. Immunohistochemistry and Immunocytochemistry

Immunohistochemistry was performed as previously described [19]. Briefly, mice were anesthetized with sodium pentobarbital (50 mg/kg, i.p.) and transcardially perfused with PBS and then with 4% paraformaldehyde in PBS. The brains were immediately removed, fixed in 4% paraformaldehyde overnight and then placed into 30% sucrose and 0.1% sodium azide in PBS, and stored at 4 °C. Brains were cut in 30 μm-thick sections on an Epredia HM 450 sliding microtome (Thermo Fisher Scientific, Rockford, IL, USA). Tissue sections were washed, blocked in 10% normal goat serum, and then incubated with anti-Iba1 (1:250, Cat #019-19741, Wako Pure Chemical Industries, Ltd., Osaka, Japan) antibody overnight at 4 °C. Following 3 washes with PBS, sections were incubated with Alexa Fluor 594 dye-conjugated secondary antibody (Life Technologies, Grand Island, NY, USA) for 1 h at room temperature in the dark. For nuclear staining, sections were incubated with 4’,6-Diamidino-2-phenylindole (DAPI) and then mounted onto the slides and coverslipped with VECTASHIELD^®^ HardSet^TM^ anti-fade mounting medium (Cat #H-1400, Vector Laboratories, Burlingame, CA, USA).

For immunocytochemistry, MMCs were grown in 96-well plates and treated with 0–5 µM of deltametrin for 24–48 h in the presence or absence of 50 µM of salubrinal. Following treatment, cells were rinsed with PBS and fixed in 4% paraformaldehyde prepared in PBS (pH 7.4). Cells were blocked in 1% BSA with 0.1% Triton X-100 for 1 h, and then incubated overnight at 4 °C with anti-Iba1 (1:500, Cat #AB019-19741, Wako Pure Chemical Industries, Ltd., Osaka, Japan), anti-gp91phox (1:250, Cat #9661, Cell Signaling, Danvers, MA, USA), anti-ATF-4 (1:250, Cat #11815, Cell Signaling, Danvers, MA, USA), and anti-CHOP (1:200, Cat #AB6199, Abcam, Cambridge, MA, USA) primary antibodies. Next, cells were incubated with Alexa Fluor 488 or 594 dye-conjugated secondary antibody (Life Technologies, Grand Island, NY, USA). For nuclear staining, a drop of ProLong Gold anti-fade mounting medium with DAPI was added into each well. All images were captured using an All-in-One Fluorescence Microscope (BZ-X810, Keyence Itasca, IL, USA).

### 2.9. Statistical Analysis

Data analysis was performed using Prism 5.01 software (GraphPad Software, San Diego, CA, USA) and presented as mean ± SEM. All *in vitro* experiments were performed three times with three replications per experiment on different days. The Student’s *t*-test with Welch’s correction was used to compare differences between two groups, and one-way ANOVA with Bonferroni’s post-hoc test was performed to compare differences among multiple groups. The *p* values < 0.05 were considered statistically significant. 

## 3. Results

### 3.1. Repeated Exposure to Deltamethrin Increases F4/80 mRNA Expression and Activates Microglia in the Hippocampus of Adult Mice

Microglial cells are the resident macrophages in the brain that proliferate and activate after neurotoxic insults. F4/80 is cell surface glycoprotein that is highly expressed in mouse microglia. Therefore, first, we investigated the effects of deltamethrin on gene expression of a mouse-specific microglial marker F4/80. We found that deltamethrin exposure significantly increased mRNA expression of F4/80 in the hippocampus (Figure 2A). The expression of F4/80 was 2.1 times higher in deltamethrin-treated mice when compared to the control group. To examine the inflammatory response of microglia after deltamethrin exposure, hippocampal microglia were visualized with Iba1 immunofluorescence staining (Figure 2B–E). In control animals, microglia in a resting condition exhibited small, round cell bodies (Figure 2B,D). Upon stimulation, microglia undergo classical morphological changes, characterized by amoeboid-shaped swollen ramified cells with a larger soma. We found an amoeboid appearance of swollen ramified microglia with larger cell bodies in the hippocampus of deltamethrin treated mice (Figure 2C,E), indicating activation of microglia caused by deltamethrin. 

### 3.2. Deltamethrin Increases Oxidative and Nitrosative Enzymes in the Mouse Hippocampus

Out of several free radical producing-enzyme systems, inducible nitric oxidesynthase (iNOS) and NADPH oxidase 2 (gp91^phox^) play major roles in response to numerous proinflammatory mediators (e.g., IL-6, TNF-α etc.) produced in microglia. Therefore, we evaluated whether exposure to deltamethrin may induce oxidative and nitrosative stress in the hippocampus through activation of gp91^phox^ and iNOS. We found that there was a significant upregulation in mRNA expression and protein levels of gp91^phox^ (Figure 3A,B) and iNOS (Figure 3C,D) in the hippocampus in deltamethrin-treated mice. The mRNA expression of gp91^phox^ was increased 1.8-fold (Figure 3A) and iNOS was increased 1.7-fold (Figure 3C). The gp91^phox^ protein was increased by 180% in deltamethrin-treated mice when compared to the corn oil control (Figure 3B). Deltamethrin increased the protein level of iNOS by 193% when compared to the control group (Figure 3D). Furthermore, we investigated the effect of deltamethrin on lipid peroxidation, as assessed by quantification of 4-HNE, a major end product of lipid peroxidation from the oxidation of omega 6-polyunsaturated fatty acids. We found a significant increase in 4-HNE in the hippocampus of deltamethin-treated mice compared to control animals (Figure 3E). These results indicate that deltamethrin may trigger oxidative and nitrosative stress in the hippocampus.

### 3.3. Deltamethrin Exposure Induces Neuroinflammation in the Mouse Hippocampus 

Activation of gp91^phox^ and iNOS in microglia can increase the release of many cytokines and chemokines [22,25,26]. Here, we assayed mRNA expressions and protein levels of TNF-α, IL-1β, and IL-6 after deltamethrin exposure. These are the key pro-inflammatory cytokines that play crucial roles in initiating and sustaining the inflammatory process. We found that deltamethrin exposure significantly increased mRNA and protein levels of TNF-α (Figure 4A,B), IL-1β (Figure 4C,D), and IL-6 (Figure 4E,F) in the hippocampus. Deltamethrin increased mRNA expression of TNF-α 2.2-fold (Figure 4A), IL-1β 1.4-fold and IL-6 1.9-fold (Figure 4C) when compared to the control group. Similarly, the protein level of hippocampal TNF-α was increased by 465% (Figure 4B), IL-1β was increased by 236% (Figure 4D), and IL-6 was increased by 150% (Figure 4F) in deltamethrin-treated mice. These data indicate that exposure to deltamethrin may induce neuroinflammation in the hippocampus of adult mice.

### 3.4. Exposure to Deltamethrin Causes Hippocampal ER Stress in Mice

To determine whether the observed neuroinflammation was associated with ER stress, we measured protein levels of ER stress markers CHOP, GRP78, and ATF-4 in the hippocampus using Western blot analysis. Exposure to deltamethrin significantly increased CHOP protein by 252% (Figure 5A), GRP78 protein by 214% (Figure 5B), and ATF-4 protein by 182% (Figure 5C) when compared to the control group. These elevations in GRP78, CHOP, and ATF-4 are indicative of activation of ER stress in the hippocampus of deltamethrin-exposed mice. 

### 3.5. ER Stress Promotes Neuroinflammation in MMCs following Deltamethrin Exposure

To determine whether the ER stress response leads to neuroinflammation after exposure to deltamethrin, an additional study was performed *in vitro* with MMCs. First, we assessed whether deltamethrin induces ER stress in MMCs by examining the ER stress-related protein CHOP in the presence or absence of salubrinal, a pharmacological inhibitor of ER stress. Deltamethrin resulted in a dose- and time-dependent increase in CHOP protein in MMCs (Figure 6). Following exposure to deltamethrin (1 and 5 µM), CHOP was increased by 204% by 1 µM and 230% by 5 µM at 24 h and by 273% and 336%, respectively, at 48 h after exposure (Figure 6A,B). Pre-treatment of cells with salubrinal prevented the upregulation of CHOP by blocking deltamethrin-induced ER stress. This result was further validated using immunofluorescence staining (Figure 6C). We also measure the protein levels of ATF-4, which were significantly increased by deltamethrin and attenuated by salubrinal co-treatment (Figure 7). To determine whether the induction of ER stress leads to neuroinflammation in MMCs, we defined the effects of deltamethrin on gene expression of inflammatory cytokines TNF-α, IL-1β, and IL-6, and analyzed whether co-treatment with salubrinal would block the upregulation (Figure 8A,B,E,F,I,J). Indeed, deltamethrin increased mRNA levels of all three cytokines at 24 h and 48 h and salubrinal blocked nearly all these inductions. To determine whether increased gene expression resulted in elevated protein levels, we further analyzed TNF-α, IL-1β, and IL-6 using the WB technique. The protein level of TNF-α was increased by 278% by 1 µM and 398% by 5 µM at 24 h and by 404% and 444%, respectively at 48 h after exposure, all of which were entirely blocked by salubrinal treatment (Figure 8C,D). Similarly, the protein levels of both IL-1β and IL-6 were also significantly increased by deltamethrin, and were attenuated by co-treatment with salubrinal (Figure 8G,H,K,L). These data indicate that neuroinflammation may be a consequence of ER stress induction following deltamethrin exposure, and that this may contribute to associated neurodegeneration [5,10].

### 3.6. Inhibition of ER Stress with Salubrinal Attenuates Deltamethrin-Induced Depletion of Antioxidant Levels and Induction of Oxidative Stress 

To determine the effects of deltamethrin on antioxidant levels, we assessed the levels of GSH MMC cells. The cellular concentration of GSH was decreased by 35 to 46% at 24 h (Figure 9A), and 55 to 80% at 48 h (Figure 9B) with 1 and 5 µM deltamethrin, respectively, when compared to the control group. Co-treatment with salubrinal significantly attenuated the deltamethrin-induced loss of cellular GSH. Deltamethrin (1 and 5 µM) increased mRNA expressions and protein levels of gp91^phox^ (Figure 10). The mRNA expression was increased 1.3 to 2.3-fold at 24 h, and 5.4 to 13.0-fold at 48 h (Figure 10A,B). Protein levels of gp91^phox^ were increased by 223–360% at 24 h and remained at the same level at 48 h after exposure (Figure 10C,D). Pre-treatment of cells with salubrinal significantly reduced deltamethrin-induced upregulation of gp91^phox^. This result was further validated using immunofluorescence staining (Figure 10E). The increased gp91^phox^ leads to increased production of ROS. Co-treatment with salubrinal significantly reduced the generation of ROS (Figure 9C,D). We also found that deltamethrin increased the level of iNOS protein in MMC cells by 165% and 224% at 24 h and by 318% and 333% at 48 h after exposure and the elevation of iNOS protein was significantly attenuated when cells were co-treated with salubrinal (Figure 9E). Together, these results demonstrate that ER stress by deltamethrin leads to neuroinflammation through disruption of antioxidant defenses and the induction of oxidative stress. 

## 4. Discussion

Chronic ER stress and neuroinflammation are critical pathological features of nearly all neurodegenerative diseases, including Alzheimer’s, Parkinson’s, and Huntington’s disease [6,27,28,29]. Recently, we revealed that exposure to deltamethrin insecticide induces ER stress [5,10] and neuroinflammation [3]. However, the mechanistic link between ER stress and neuroinflammation has not been revealed. Here, we demonstrate that deltamethrin may induce neuroinflammation through ER stress pathways. Our results show that one month of repeated exposure to 3 mg/kg of deltamethrin causes microglial activation, oxidative/nitrosative stress and crucially increases the major pro-inflammatory cytokines in the hippocampus, indicating induction of potential neuroinflammation in the hippocampus. These effects were preceded by increased levels of CHOP, ATF-4, and GRP78 proteins. Using MMC line, we further found that pharmacological inhibition of ER stress with salubrinal robustly attenuated the levels of GSH, 4-HNE, gp91^phox^, iNOS, ROS, IL-1β, TNF-α, and IL-6, suggesting that induction of ER stress may trigger the inflammatory response following deltamethrin exposures.

Environmental exposure to pesticides has been associated with an increased risk of neurodegenerative diseases [30,31,32]. Chronic ER stress has been identified as a potential contributor to neurodegeneration associated with these diseases [2,33,34]. Upregulation of GRP78 and CHOP are characteristic indicators of ER stress as these proteins are ubiquitously expressed at very low levels under nonstress conditions but are robustly activated when cells experience ER stress [35,36]. Likewise, in our previous studies, sub-chronic exposure to deltamethrin resulted in ER stress as GRP78 and CHOP proteins were significantly increased in the hippocampus [10]. Studies show that phosphorylation of eIF2α activates translational induction of ATF-4, which in turn causes activation of CHOP during ER stress [37,38]. Similarly, we found deltamethrin exposure upregulated ATF-4 that correlated with increased levels of CHOP in MMCs. Research has shown that ATF-4−/− cells fail to induce CHOP during ER stress [39]. A growing body of evidence indicates that ER stress can elicit neuroinflammation through the activation of multiple intracellular signaling pathways, including NF-_k_B, NADPH oxidase, iNOS, and through the production of ROS, and proinflammatory cytokines [6,28,40,41,42]. Thus, deltamethrin exposure may lead to neuroinflammation in the hippocampus through ER stress pathways in mice. 

Human post-mortem brain tissues and animal models of PD, AD, and HD consistently exhibit the presence of activated microglia, and the increased expression of pro-inflammatory and neurotoxic factors, including TNF-α, IL-6, iNOS, and gp91^phox^ in the striatum and hippocampus, which may exacerbate neuronal loss in these brain areas [43,44,45,46,47]. Here, we report that deltamethrin exposure causes microglial activation and increases mRNA and protein levels of TNF-α, IL-1β, IL-6, iNOS, 4-NHE, and gp91^phox^ in the hippocampus, indicating induction of potential oxidative stress and neuroinflammation in the hippocampus. Activation of microglia and secretion of inflammatory cytokines in the hippocampus and cortex have been reported in ischemic brains [48]. The activation of iNOS generates nitric oxide (NO), which interacts with superoxide and forms highly toxic reactive radical peroxynitrite, which in turn leads to neuroinflammation and neuronal death during ischemia [49,50], and the neurodegeneration in AD and PD [51,52,53]. Gp91^phox^ is the major subunit of NADPH oxidase accountable for the generation of ROS and superoxide, which have been shown to be significant contributors to neurodegeneration [54,55,56]. Exposure to 1-methyl-4-phenyl-1,2,3,6-tetrahydropyridine (MPTP) has been shown to cause activation of microglia and gp91^phox^ in the mouse brain [46,57,58,59]. Further, mice lacking gp91^phox^ exhibit reduced microglial activation and neurodegeneration after a traumatic brain injury or exposure to lipopolysaccharide (LPS) and MPTP [25,57,60,61]. 

To identify whether deltamethrin-induced neuroinflammation occurred through ER stress pathways, additional experiments were performed using the MMC line. Microglia are phagocytic cells that respond to inflammatory stimuli, such as environmental toxicants, and become activated to repair and protect the resident neurons [62,63,64,65]. However, continued overactivation of microglia can lead to chronic and sustained neuroinflammation and result in more neuronal cell death, leading to neurodegenerative diseases [64,65,66]. Here, we found that exposure to deltamethrin decreased GSH and increased gp91^phox^_,_ iNOS, 4HNE, ROS, and inflammatory cytokines TNF-α, IL-1β, and IL-6 in MMCs, which were correlated with increased levels of CHOP, ATF-4, and GPRP78. During the initiation of the ER stress, CHOP is induced by the transcription factor eIF2α, which is phosphorylated by protein kinase R (PKR) and halts the translation of proteins, thereby decreasing ER stress in the cell [8,18]. In this study, we noticed that the inhibition of eIF2α with salubrinal abolished elevated levels of ATF-4 and CHOP and restored the levels of GSH, and prevented activation of gp91^phox^, iNOS, and the production of ROS and inflammatory cytokines in MMCs, indicating the involvement of ER stress pathways in neuroinflammation after deltamethrin exposure. 

When antioxidant defense mechanisms fail to effectively counter the sources of ROS, oxidative stress and damage occur within the cells. Glutathione (GSH) is a potent antioxidant and detoxifying agent in the brain that provides protection from oxidative stress-induced damage through the reduction of ROS and the maintenance of redox homeostasis in brain [67,68]. Deltamethrin exposure significantly reduced intracellular levels of GSH in MMCs that were attenuated by inhibition of ER stress with salubrinal. The reduction in GSH reflects a reduced antioxidant capacity of cells, which may lead to increased ROS and elicits oxidative stress and neuroinflammation in the hippocampus. In the mammalian brain, glial cells of the cortex and hippocampus have been shown to have the highest concentration of GSH [69]. Recent studies show that GSH levels decrease in the hippocampus and cortex with age and age-related neurological disorders, and are found to be correlated with a decline in cognitive function [69,70]. Further, the production of 4-HNE by lipid peroxidation has been associated with oxidative stress-induced neuroinflammation in aging and neurodegenerative diseases [71,72]. Deltamethrin exposure significantly increased protein levels of 4-HNE in MMCs, and was correlated with the production of ROS and inflammatory cytokines. Several studies have shown that 4-HNE acts as a critical signaling molecule that triggers inflammatory responses after the induction of oxidative stress in microglia and astrocytes [73,74,75]. These results suggest that an imbalance in antioxidant homeostasis and oxidative stress triggers an inflammatory response through ER stress pathways, following exposure to deltamethrin. 

In addition, ER stress may activate nuclear factor-κB (NF-κB), which leads to neuroinflammation flowing exposure to deltamethrin [6,28]. NF-κB is a central transcription factor that regulates a large array of genes involved in inflammatory responses [76,77,78]. Recent evidence indicates that NF-κB can be activated by UPR during ER stress [6,28,77,79]. Thus, the induction of ER stress by deltamethrin may cause activation of NF-κB, which leads to neuroinflammation. Further studies are warranted to address this hypothesis.

In summary, our data demonstrate that deltamethrin may contribute to neuroinflammation in the hippocampus through the activation of microglia, leading to oxidative/nitrosative stress and the release of pro-inflammatory cytokines through the activation of ER stress pathways. Furthermore, inhibition of ER stress pathways by salubrinal greatly prevents deltamethrin-induced decreased levels of GSH and increased levels of TNF-α, IL-1β, IL-6, gp91^phox^, ROS, and iNOS in deltamethrin-treated MMCs, suggesting that the neuroinflammatory response is probably a secondary effect from ER stress after exposure to deltamethrin in mice. Additional studies are underway to elucidate the precise role of ER stress in neuroinflammation following exposure to deltamethrin; these studies take a genetic approach, which might provide a novel mechanistic pathway leading to a potential therapeutic intervention to inhibit the neuroinflammation that is the root cause of neurodegenerative diseases.

## Figures and Tables

**Figure 1 cells-11-01961-f001:**
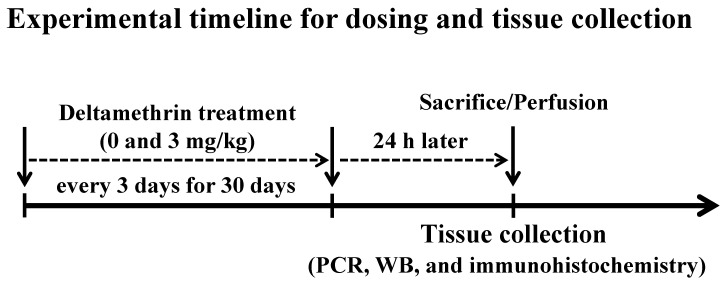
Mice were given 0 (corn oil) or 3 mg/kg of deltamethrin by oral gavage every three days for thirty (30) days. Twenty-four hours after the last dose, one set of animals (*n* = 5/group) were sacrificed and hippocampi were collected for gene and protein analysis, and another set of animals (*n* = 5/group) were perfused for immunohistochemical study.

**Figure 2 cells-11-01961-f002:**
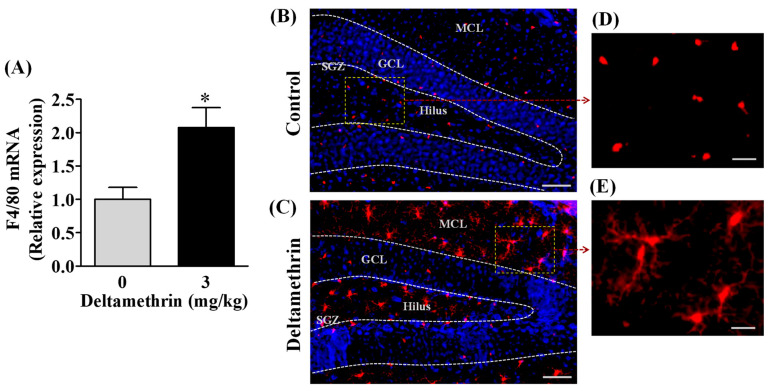
Repeated exposure to deltamethrin induces microglial activation and increases F4/80 mRNA in the hippocampus of adult mice. (**A**) mRNA expression of hippocampal F4/80. Asterisk denotes significant difference from control (*p* < 0.05). (**B**) Microglia in resting condition in corn oil control (top panel), and (**C**) activated microglia in deltamethrin-treated mice (bottom panel). GCL, granule cell layer; SGZ, sub granular zone; MCL, molecular cell layer. Scale bar = 400 μm. Higher magnification images show representative Iba1+ cells from control animals (**D**) and deltamethrin-treated mice (**E**). Scale bar = 50 μm.

**Figure 3 cells-11-01961-f003:**
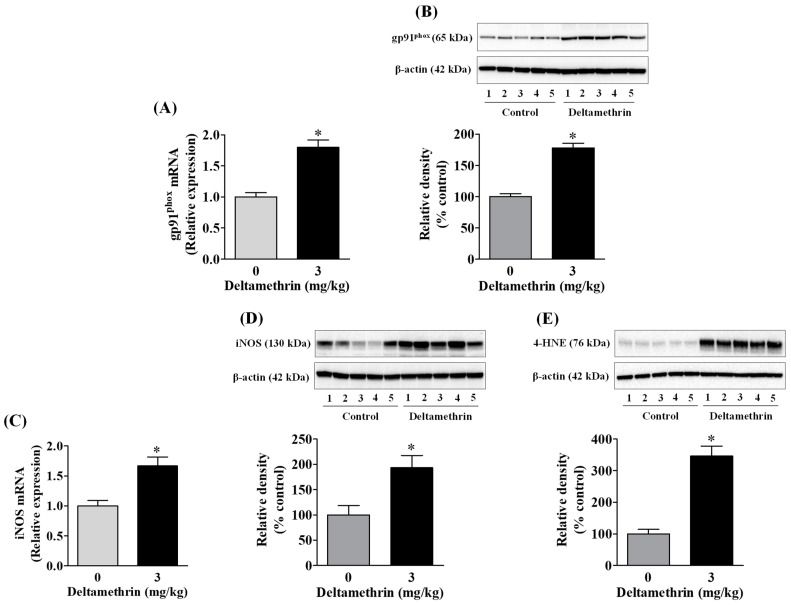
Deltamethrin exposure increases oxidative and nitrosative enzymes in the hippocampus. (**A**) mRNA and (**B**) protein levels of gp91^phox^, (**C**) mRNA and (**D**) protein levels of iNOS, and (**E**) protein levels of 4-HNE. The protein bands were quantified by densitometry from Western blot analysis, and the results reported as relative band densities in bar graphs. An asterisk denotes significant difference from the control (*p* < 0.05).

**Figure 4 cells-11-01961-f004:**
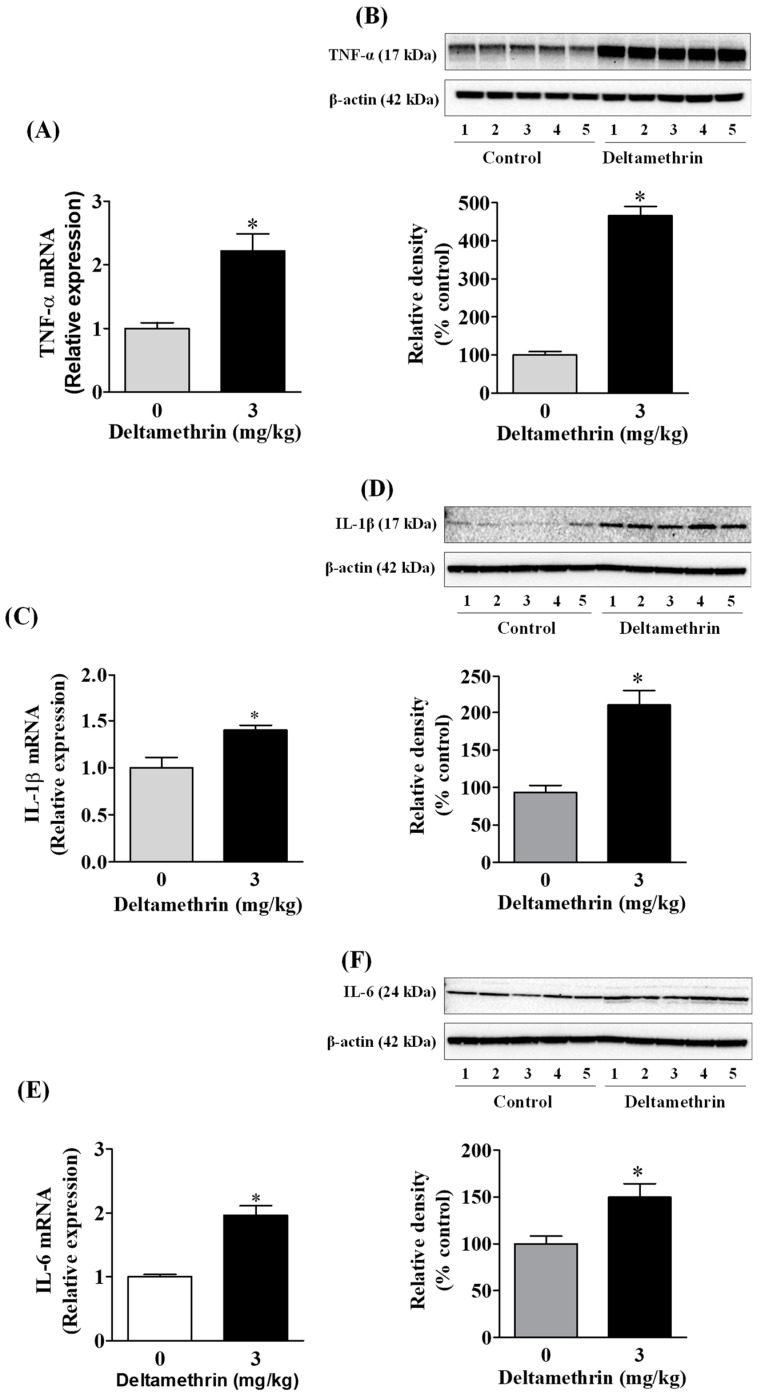
Deltamethrin induces the inflammatory cytokines in the adult mouse hippocampus. (**A**) mRNA and (**B**) protein levels of TNF-α, (**C**) mRNA and (**D**) protein levels of IL-1β, and (**E**) mRNA and (**F**) protein levels of IL-6. Since the membrane was stripped and re-probed with anti-IL-6 after taking the image of IL-1β, the same loading control is used for both proteins. The protein bands were quantified by densitometry from Western blot, and the results reported as relative band densities in bar graphs. An asterisk denotes significant difference from control (*p* < 0.05).

**Figure 5 cells-11-01961-f005:**
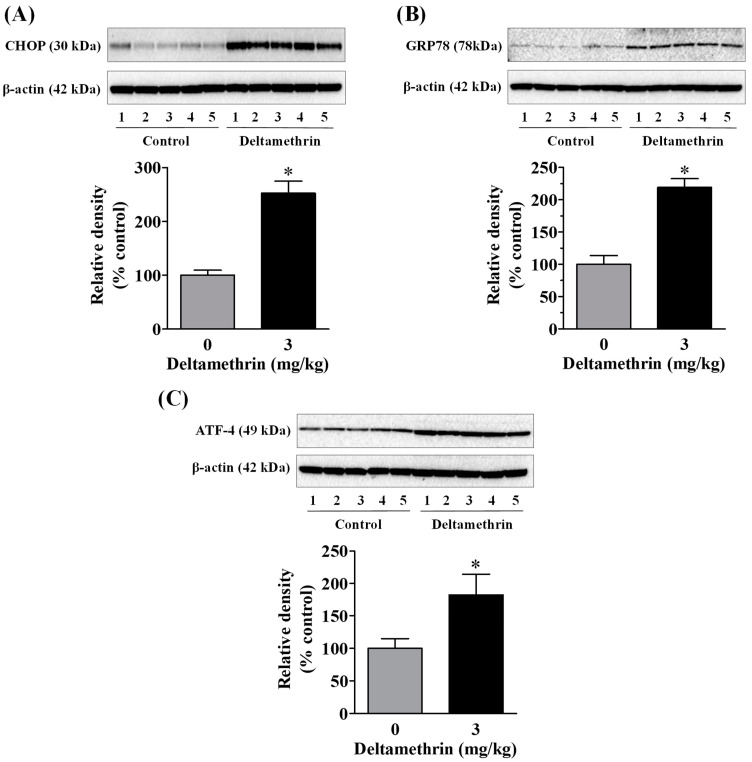
Deltamethrin exposure causes hippocampal ER stress in mice. Protein levels of CHOP (**A**), GRP78 (**B**), and (**C**) ATF-4 were determined by Western blotting. The protein bands were quantified by densitometry from Western blot analysis and the results reported as relative band densities in bar graphs. An asterisk denotes significant difference from control animals (*p* < 0.05).

**Figure 6 cells-11-01961-f006:**
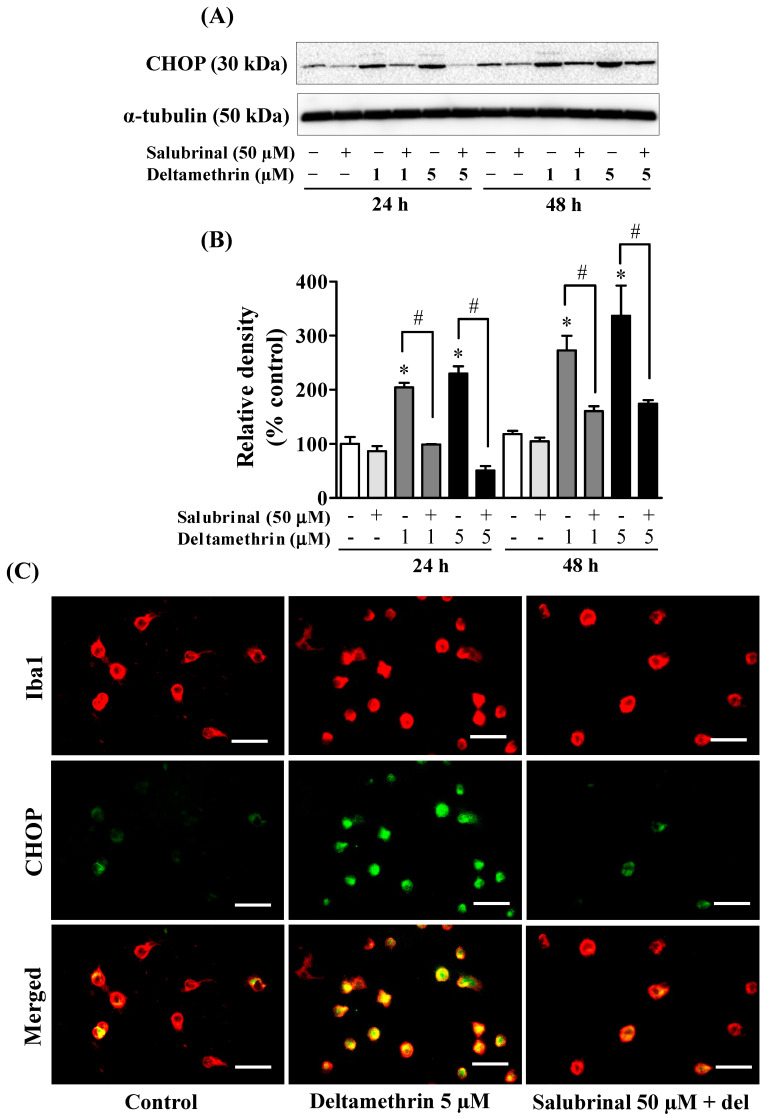
Salubrinal prevents deltamethrin-induced activation of stress pathways in MMCs cells. Protein levels of CHOP in MMCs were quantified by densitometry from Western blots (**A**). The results are reported as relative band densities in bar graphs (**B**). The values represent mean ± SEM from three independent experiments, each of which was performed in triplicate. An asterisk denotes significant difference from the control group, and a hashtag denotes significant differences between deltamethrin and deltamethrin + salubrinal (*p* < 0.05). Expression of CHOP in MMCs (**C**). MMCs were labeled with anti-Iba1 antibody (red) and CHOP protein was visualized by immunolabeling (green). Co-localization of CHOP and Iba1 is shown in yellow. Scale bar = 200 µm.

**Figure 7 cells-11-01961-f007:**
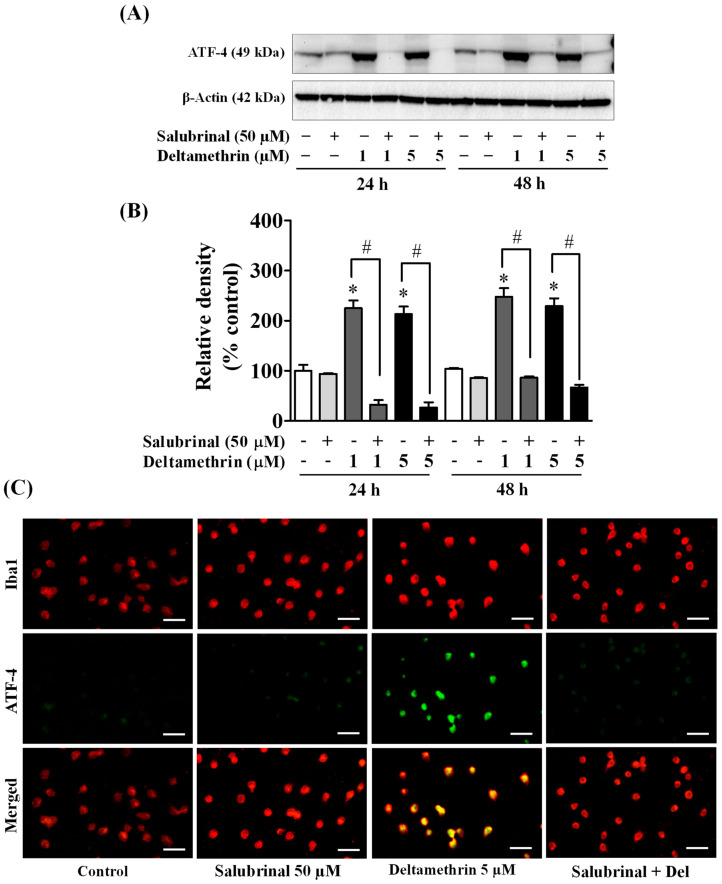
Salubrinal prevents deltamethrin-induced increased levels of ATF-4 in MMCs. Protein levels of ATF-4 in MMCs were quantified by densitometry from Western blots (**A**). The results are reported as relative band densities from representative blots in bar graphs (**B**). The values represent mean ± SEM from three independent experiments, each of which was performed in triplicate. An asterisk denotes significant difference from the control group, and a hashtag denotes significant differences between deltamethrin and deltamethrin + salubrinal (*p* < 0.05). Expression of ATF-4 protein in MMCs was visualized by immunofluorescence staining (**C**). MMCs were labeled with anti-Iba1 antibody (red). ATF-4 protein was visualized by immunolabeling (green). Co-localization of ATF-4 and Iba1 is shown in yellow. Scale bar = 200 µm.

**Figure 8 cells-11-01961-f008:**
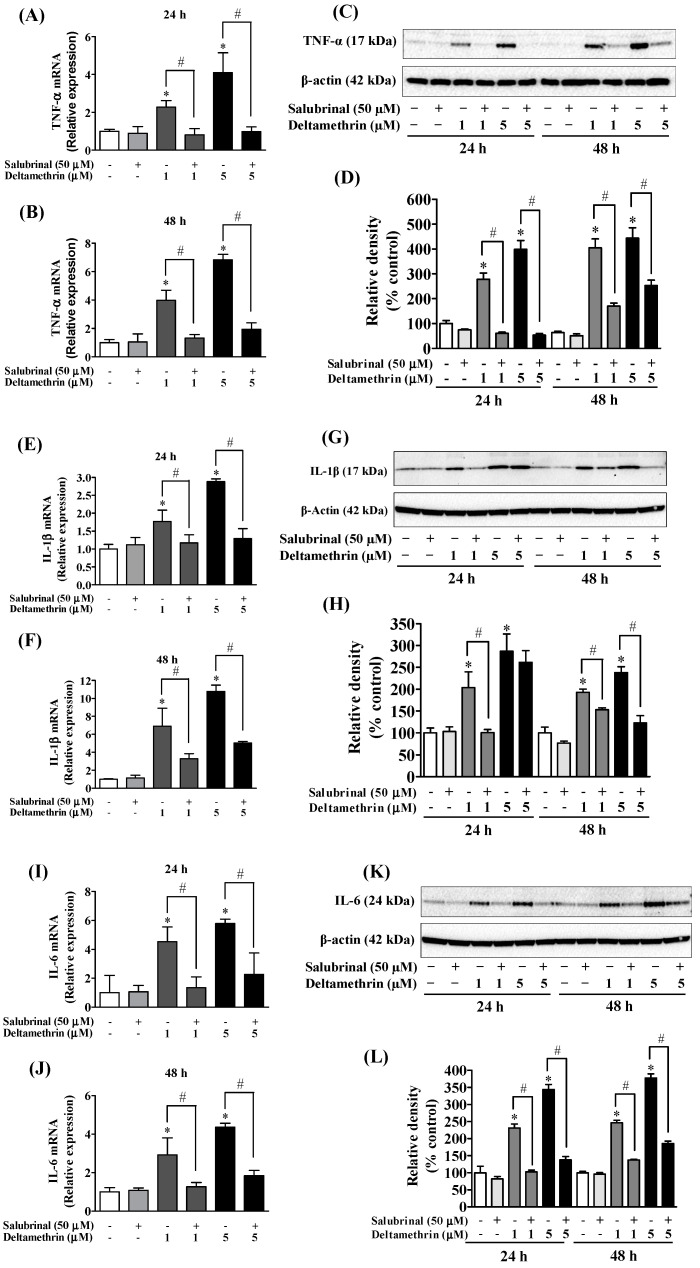
Salubrinal reduces deltamethrin-induced increased levels of inflammatory cytokines in MMCs. (**A**,**B**) mRNA and (**C**,**D**) protein levels of TNF-α; (**E**,**F**) mRNA and (**G**,**H**) protein levels of IL-1β; (**I**,**J**) mRNA and (**K**,**L**) protein levels of IL-6. The protein bands were quantified by densitometry and the results reported as relative band densities. As the membrane was stripped and re-probed with anti-IL-6 after IL-1β, the same loading control was used for both proteins. The values represent mean ± SEM from three independent experiments, each of which was performed in triplicate. An asterisk denotes significant difference from the control group, and a hashtag denotes significant differences between deltamethrin and deltamethrin + salubrinal (*p* < 0.05).

**Figure 9 cells-11-01961-f009:**
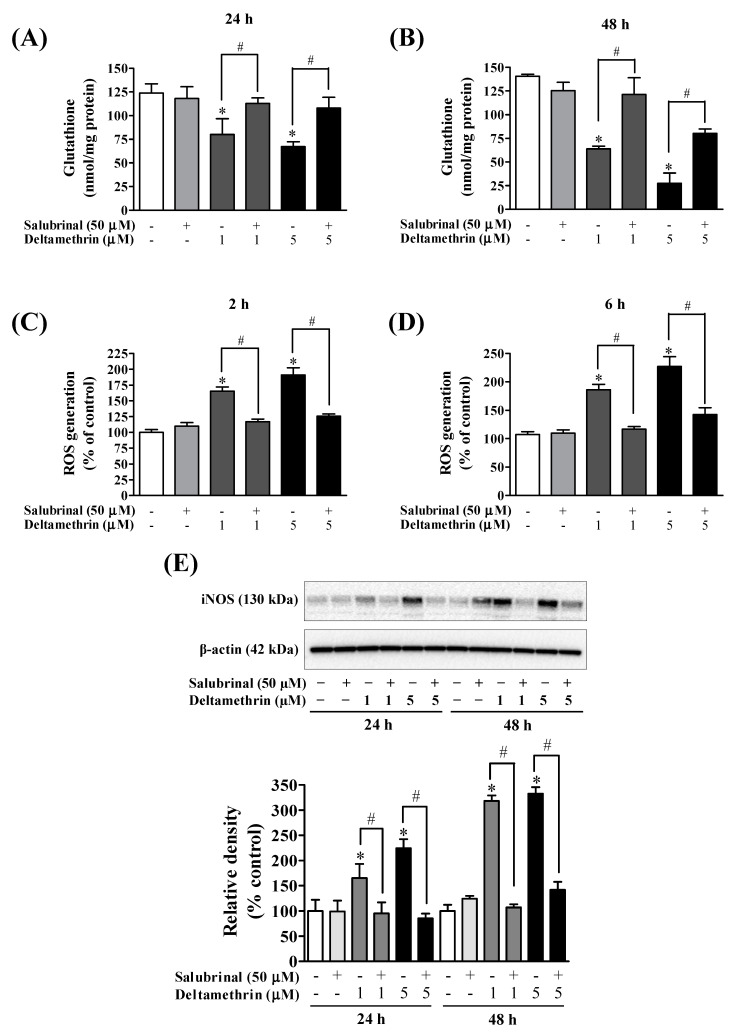
Salubrinal attenuated deltamethrin-induced decreased levels of GSH and increased levels of iNOS and ROS production. (**A**,**B**) levels of intracellular GSH, (**C**,**D**) generation of ROS, (**E**) protein levels of iNOS in MMCs cells. The protein bands were quantified by densitometry and the results reported as relative band densities. The values represent mean ± SEM from three independent experiments, each of which was performed in triplicate. An asterisk denotes significant difference from the control group, and a hashtag denotes significant differences between deltamethrin and deltamethrin + salubrinal (*p* < 0.05).

**Figure 10 cells-11-01961-f010:**
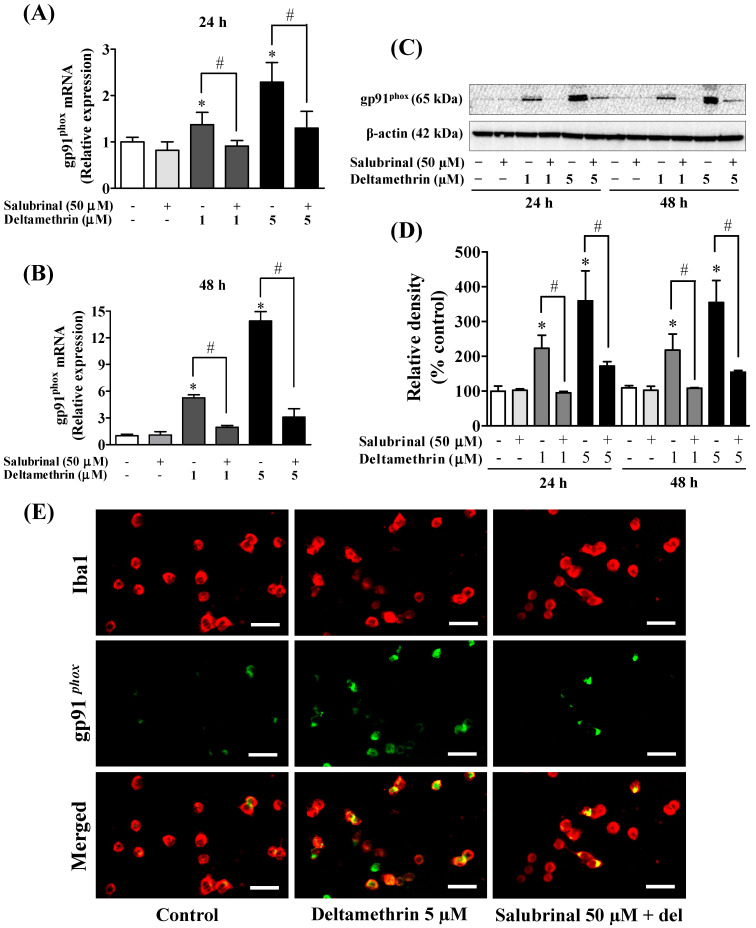
Inhibition of ER stress factor eIF2α with salubrinal reduces deltamethrin-induced increased levels of gp91^phox^ in MMCs cells. (**A**,**B**) mRNA and (**C**,**D**) protein levels of gp91^phox^. The protein bands were quantified by densitometry and the results reported as relative band densities. Since the membrane was stripped and re-probed with anti-gp91^phox^ after ATF-4, the same loading control is used from Figure 7. An asterisk denotes statistically significant difference from the control group, and a hashtag denotes statistically significant differences between deltamethrin and deltamethrin + salubrinal (*p* < 0.05). (**E**) Expression of gp91^phox^ protein in MMCs was visualized by immunofluorescence staining. MMCs were labeled with anti-Iba1 antibody (red). Expression of gp91^phox^ protein was visualized by immunolabeling (green). Co-localization of gp91^phox^ and Iba1 is shown in yellow. Scale bar = 200 µm.

**Table 1 cells-11-01961-t001:** Mouse primer sequences for qPCR.

Primers	Forward	Reverse
F4/80	5′-TCCAGCACATCCAGCCAAAGCA-3′	5′-CGTAAGCTCCCCAGAAGCCATGA-3′
TNF-α	5′-TAGCCCACGTCGTAGCAAA-3′	5′-CCTTGAAGAGAACCTGGGAGT-3′
IL-1β	5′-CTTGTGCAAGTGTCTGAAGCA-3′	5′-TGCGAGATTTGAAGCTGGATG-3′
IL-6	5′-TCCTCTCTGCAAGAGACTTCC-3′	5′-TCCACGATTTCCCAGAGAACA-3′
gp91phox	5′-TCCTGCTGCCAGTGTGTCGAAA-3′	5′-TGCAATTGTGTGGATGGCGGTG-3′
iNOS	5′-CCAGTGCCCTGCTTTGTGCG-3′	5′-TGCAATTGTGTGGATGGCGGTG-3′
GAPDH	5′-GGGCTGGCATTGCTCTCAATGAC-3′	5′-TCTTGCTCAGTGTCCTTGCTGGG-3′

## Data Availability

The datasets generated and/or analyzed during the current study are available from the corresponding author upon reasonable request.

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
