# Peer review of "Deltamethrin-Evoked ER Stress Promotes Neuroinflammation in the Adult Mouse Hippocampus"

_cells, 2022, doi:10.3390/cells11121961_

Round 1
Reviewer 1 Report
This paper focuses on hippocampal ER stress induced by deltamethrin. The topic is of interest in the field of neurotoxicology and has possible public health implications related to neurological disease. This is a very well conducted study and well written manuscript. There is a single overarching suggestion to increase impact.
Major concerns:
1. A few other brain regions should be analyzed to determine whether the hippocampal pathology is specific or brain-wide (at minimum in Figure 1).
Author Response
We would like to thank you for taking the time and effort necessary to review this manuscript and sincerely appreciate your comments and suggestions. The manuscript has been extensively revised for grammar and spellings.
- A few other brain regions should be analyzed to determine whether the hippocampal pathology is specific or brain-wide (at minimum in Figure 1).
Response: Since we previously found that the hippocampus is particularly more susceptible to ER stress and apoptosis after exposure to deltamethrin in adult mice (Hossain et al., 2019), we have investigated oxidative stress and neuroinflammation in the hippocampus in this study.
However, according to your suggestion, we have examined an ER stress marker CHOP in the cortex and striatum within the given timeframe. We found that the hippocampus is more sensitive to deltamethrin as evidenced by significantly increased levels of CHOP in the hippocampus (252%) (Figure 5A in the revised manuscript) than the frontal cortex (25%) (Figure below). No significant changes in the levels of CHOP were observed in the striatum. We have not included these data in this manuscript.
As you suggested, we will rigorously investigate other brain regions such as cerebellum, cortex, and striatum to precisely determine regional susceptibility in our future study.

Reviewer 2 Report
The work by Hossain et al has significant significance to the research in its field. But this manuscript is not presented with proper controls and the work requires sincere introspection before thinking to process this for publication. Some major concerns are:
1. What was used in the treatment for control mice? From the manuscript it is not clear and it appears the authors are using untreated control whereas it should have been treated with corn oil! -This needs proper clarification.
2. Many western blots like the ones represented in Fig 3B,E; Fig 5B; Fig 6A; Fig 9E shows two bands contradicting the major available published literature for the reported proteins. The authors should present full uncropped unprocessed raw western blot data for all the images reported in the manuscript (along with their technical/biological repeats used for statistics estimation) indicating the molecular weight marker and proper labels. What steps were taken to validate the antibodies used by the authors to test their specificity in the given system?
3. In Fig2a why did the authors cherry pick just F480 expression to showcase the role of macrophages? What happens to the neutrophils, monocyte and basophils? Findings of line # 215-217 seems very random and is not well defined to give a mechanistic point of evidence.
4. This is in respect to Fig2 in vivo work, the authors should have used proper controls like have they ever compared the effect of known ER stressors on microglial morphology? That would be vital to propose the idea about deltamethrin mediated ER stress. Ideally salubrinal +ER stressor (like Thapsigargin etc) should show reduction in microglial activation and should have been compared with the effect of Deltamethrin+salubrinal to conclude the ER stress mechanism.
Author Response
We would like to thank you for taking the time and effort necessary to review this manuscript. We also sincerely appreciate your comments and suggestions, which helped us to improve the quality of our manuscript. The manuscript has been extensively revised for grammar and spellings.
- What was used in the treatment for control mice? From the manuscript it is not clear and it appears the authors are using untreated control whereas it should have been treated with corn oil! -This needs proper clarification.
Response: Control animals were treated with corn oil alone. To make it very clear, we have revised the sentence as “Animals in control group were given corn oil alone and animals in treatment group were given 3 mg/kg of deltamethrin in corn oil” (Page: 3, Lines: 98-99).
- Many western blots like the ones represented in Fig 3B,E; Fig 5B; Fig 6A; Fig 9E shows two bands contradicting the major available published literature for the reported proteins. The authors should present full uncropped unprocessed raw western blot data for all the images reported in the manuscript (along with their technical/biological repeats used for statistics estimation) indicating the molecular weight marker and proper labels. What steps were taken to validate the antibodies used by the authors to test their specificity in the given system?
Response: Sorry, some western blots had two bands. We have rerun the western blots and have fixed these figures in our revised manuscript. We found that there were two bands with these western blots before because of improper blocking, and inadequate washing. The full uncropped unprocessed raw western blot images are provided with revised manuscript in a PDF file.
All statistical analyses were performed on raw data. All in vivo data represent mean ± SEM from 5 animals per group. All in vitro data present mean ± SEM from 3 individual experiments preformed on different days. This information is available in method section under statistical analysis.
To validate the specificity of antibodies, tissue section and cells and WB membrane were incubated in the presence or absence of primary antibodies. There was no band/signal when a membrane or tissue section/cells were incubated without primary antibody. We used antibody specific bands that also correlated with animal tissues and cell lysates.
- In Fig2a why did the authors cherry pick just F480 expression to showcase the role of macrophages? What happens to the neutrophils, monocyte and basophils? Findings of line # 215-217 seems very random and is not well defined to give a mechanistic point of evidence.
Response: Microglial cells are the resident macrophages in the brain that proliferate and activated after neurotoxic insults. F4/80 is the cell surface glycoprotein that is highly expressed in mouse microglia. Therefore, we investigated the effects of deltamethrin on gene expression of a mouse-specific microglial marker F4/80.
We don’t know what happened to the neutrophils, monocyte and basophils as we never examined these cells. Since activation of microglia results in neuroinflammation and neuronal damages, we just focused on microglia in this study.
We have revised the sentence accordingly (Lines: 242-245). The statement is now well defined and provide a mechanistic point of evidence.
- This is in respect to Fig2 in vivo work, the authors should have used proper controls like have they ever compared the effect of known ER stressors on microglial morphology? That would be vital to propose the idea about deltamethrin mediated ER stress. Ideally salubrinal +ER stressor (like Thapsigargin etc) should show reduction in microglial activation and should have been compared with the effect of Deltamethrin+salubrinal to conclude the ER stress mechanism.
Responses: Thank you for your valuable comments. In this study, we visualized microglia with Iba1 antibody to see whether deltamethrin results in microglial activation. Since we did not treat the animals with Salubrinal + deltamethrin or known ER stressors such as Thapsigargin, we were not able to compare microglial morphology at this time.
According to your suggestion, in our future studies, we will treat a cohort of animals with salubrinal + deltamerthin and salubrinal + known ER stressor Thapsigargin to compare microglial morphology and precisely conclude the role of ER stress in neuroinflammation in vivo.

Round 2
Reviewer 2 Report
Issues with technical problem needs to be fixed before the manuscript can be considered for publication any longer. In one figure panel namely Fig5b, GRP78 is single band then going back to Fig4b its a double band. This is bizarre and beyond any logical explanation. These western blots needs to be properly done and replaced. I still donot see these raw data for the two mentioned conflicting bands in the suggested two figs. Its hard to interpret data when you are not sure which band to trust. The raw data doesnot indicate the figure number, molecular weight labels and doesnot represent all "repeats" (which was used for estimating statistics) for all the western data.
Author Response
Response:
It seems that you overlooked the Figs 4B and 5B. None of these figures have double bands. Furthermore, the Fig 4B represents a western blot associated with TNF-α where Fig 5B represents a western blot associated with GRP78.
We already reran the western blot for GRP78 and replaced the figure with the new image which can be seen in Fig 5B in our revised manuscript. Therefore, in Fig 5B, there is only a single band, which is the correct image.
Since we were asked to keep the track changes, both images are seen in the manuscript version that contains the viewed edits made to the manuscript. If the manuscript is viewed with the accepted edits, it can be seen that in Fig 5B, GRP78 only has 1 band present, the previous figure was removed and replaced, and Fig 4B does not contain any GRP78 results.
TNF-α Raw Data for Figure 4B
Animal Number |
Control |
Deltamethrin |
1 |
253.911 |
1744.658 |
2 |
347.985 |
1610.392 |
3 |
424.045 |
1686.727 |
4 |
454.228 |
2118.984 |
5 |
406.872 |
1622.39 |
GRP78 Raw Data for Figure 5B
Animal Number |
Control |
Deltamethrin |
1 |
47.67 |
100.35 |
2 |
37.83 |
106.45 |
3 |
29.29 |
111.47 |
4 |
66.04 |
76.32 |
5 |
46.43 |
103.96 |

Round 3
Reviewer 2 Report
The authors have properly clarified all my queries. The manuscript looks good for publication.